# Longitudinal Outcomes of Gender Identity in Children (LOGIC): study protocol for a retrospective analysis of the characteristics and outcomes of children referred to specialist gender services in the UK and the Netherlands

Eilis Kennedy [1,2,3] Chloe Lane,[1] Hannah Stynes,[1] Veronica Ranieri [1,2] Lauren Spinner,[1] Polly Carmichael,[4] Rumana Omar,[5] Victoria Vickerstaff,[6,7] Rachael Hunter [6] Robert Senior,[1,3] Gary Butler,[4,8] Simon Baron-Cohen,[9] Nastasja de Graaf,[10,11] Thomas D Steensma,[10,11] Annelou de Vries,[10,12] Bridget Young,[13] Michael King[7]

MK deceased.

For numbered affiliations see end of article.

**Correspondence to**
Dr Eilis Kennedy;
ekennedy@tavi-port.nhs.uk

## ABSTRACT

**Introduction** Specialist gender services for children and young people (CYP) worldwide have experienced a significant increase in referrals in recent years. As rates of referrals increase, it is important to understand the characteristics and profile of CYP attending these services in order to inform treatment pathways and to ensure optimal outcomes.

**Methods and analysis** A retrospective observational study of clinical health records from specialist gender services for CYP in the UK and the Netherlands. The retrospective analysis will examine routinely collected clinical and outcome measures data including demographic, clinical, gender identity-related and healthcare resource use information. Data will be reported for each service and also compared between services. This study forms part of a wider programme of research investigating outcomes of gender identity in children (the Longitudinal Outcomes of Gender Identity in Children study).

**Ethics and dissemination** The proposed study has been approved by the Health Research Authority and London—Hampstead Research Ethics Committee as application 19/LO/0181. The study findings will be published in peer-reviewed journals and presented at both conferences and stakeholder events.

### Strengths and limitations of this study

► This study will involve a retrospective analysis of routinely collected data from two European specialist gender services in a large cohort of children and young people (CYP) aged ≤13 years.

► A detailed evaluation of service use and costs will be ascertained for specialist gender services for CYP in both the UK and the Netherlands.

► As data from two distinct specialist gender services (in the UK and the Netherlands) will be used, not all variables will be available for CYP from both services.

► As the study will use data extracted from clinical health records, there will inevitably be some missing data but this will be taken into account in the analyses.

► Selection of measures and variables is constrained to those that are routinely collected by the services but limitations of these will be discussed when interpreting the findings.

## INTRODUCTION

In recent years, specialist gender services for children and young people (CYP) have experienced a significant rise in referrals worldwide.[1 2] For example, the gender identity development service (GIDS) at the Tavistock and Portman National Health Service (NHS) Foundation Trust (NHSFT) in the UK has reported a 382% increase from 678 referrals in 2014/2015 to 2590 referrals in 2018/2019.[3] Time trends in relation to the profile of CYP referred to services have also been noted. In particular, there has been a shift in recent years to an increase in referrals of CYP assigned female at birth.[4 5] However, a recent study of time trends in adolescent referrals in the Netherlands found that, other than a shift in sex ratio, no other time trends were observed in relation to demographics or intensity of gender dysphoria.[6] As rates of referrals increase, it is important to understand the characteristics and profile of CYP

attending these services in order to inform treatment pathways and to ensure optimal outcomes for these CYP.

The specialist gender services at the Tavistock and Portman NHSFT in the UK and Amsterdam University Medical Centre (AUMC) in the Netherlands are two of the longest established and largest services in Europe for CYP seeking support in relation to their gender identity. Both of the services follow a similar assessment protocol and both have consistently used the same outcome measures over the past 8 years, enabling comparisons to be made between the services. These include a multidisciplinary clinical assessment and completion of the Child Behaviour Checklist (CBCL), the Youth Self Report (YSR) and the Teacher Report Form (TRF)[7] to assess emotional and behavioural functioning, and the Social Responsiveness Scale (SRS)[8] to assess autistic traits. Data from these services have previously been compared and some cross-cultural differences in the nature of referrals to these clinics have previously been reported. For example, a recent study comparing psychological functioning in adolescents (aged 12–18 years) referred to specialist gender services across four European countries (including the UK and the Netherlands) identified that, at the time of referral, emotional and behavioural problems and peer relationship difficulties were most prevalent in CYP presenting to services in the UK.[9] Conversely, these issues were least prevalent in CYP presenting to services in the Netherlands. In addition, a greater number of younger children presenting to specialist gender services who have already made changes in their clothing, hairstyle, first name and pronouns to reflect their gender identity (sometimes referred to as social transition) at the time of referral has been reported.[10] Recent research has suggested that making these changes early on can have desirable outcomes for CYP.[11–13] However, research on this matter in prepubertal children is limited, and thus, we need to know more about how making such changes in dress and behaviour relates to later outcomes.[14] This study will extend current understanding of the nature of CYP referrals, particularly in relation to younger children, by providing further opportunity to characterise the profile and outcomes of attendees at each service, while also exploring potential cross-cultural differences.

Several longitudinal prospective cohort studies of CYP attending specialist gender services are now ongoing.[15 16] However, retrospective analysis of clinical data can provide important and unique insights into the characteristics and outcomes of CYP referred to these services which are not yet available from these ongoing prospective studies. This is particularly pertinent as it is widely acknowledged that the evidence base on which current treatment protocols is based is limited.[14] Furthermore, retrospective studies include a whole specified cohort, which is not necessarily feasible within prospective studies which generally require a process of recruitment and novel data collection. Retrospective studies of clinical cohorts, therefore, provide a valuable and informative addition to the literature.

As much of the existing literature has focused on adolescents, little is currently known about the overall characteristics and outcomes of younger CYP, particularly prepubertal and early pubertal children, who attend specialist gender services. Consequently, there is limited evidence to inform the likely trajectories and outcomes of these CYP and to enable clinical care pathways to be tailored accordingly. The present study aims to address this gap in the literature by profiling CYP aged ≤13 years who first attended specialist gender services across an 8-year period (2009–2017). It will describe and, where possible, compare the outcomes of CYP attendees in relation to their demographic and family backgrounds, emotional and behavioural functioning, autistic traits and gender identity (eg, diagnosis of gender dysphoria and social transition). Measures relating to emotional and behavioural functioning and autistic traits will be included, as mental health conditions and autism have been reported to co-occur for some CYP who are referred to specialist gender services.[17–20] Differences in referrals and treatment pathways in each country will be explored, for example, numbers of children presenting to the clinic who have already socially transitioned and age at time of referral to paediatric endocrinology. Service use and outcomes will be identified, particularly in relation to CYP who attend a paediatric endocrinology clinic and those who do not. A healthcare resource costing of both services will also be undertaken. It is anticipated that the proposed research will improve understanding of the characteristics of service users in order to help in the planning and organisation of services and to address the need for tailored support when required.

## Aims

This study aims to identify (1) the profile of CYP aged ≤13 years attending specialist gender services in the UK and the Netherlands between 2009 and 2017 and (2) the proportion of these CYP who (A) experience gender dysphoria, (B) socially transition, (C) access medical treatment (eg, hormone blockers and cross sex hormones) and (D) have co-occurring autistic traits; (3) the profile of CYP who attend paediatric endocrinology clinics and the profile of CYP who do not attend these clinics; (4) the service use and costs of CYP attending specialist gender services; (5) costs for CYP who attend paediatric endocrinology clinics and costs for CYP who do not attend these clinics.

## METHODS AND ANALYSIS
### Study design

A retrospective observational study of clinical health records from specialist gender services in the UK and the Netherlands across an 8-year period (2009–2017). See table 1 for a full list of variables and measures. This study forms part of a wider programme of research investigating outcomes of gender identity in children (the Longitudinal Outcomes of Gender Identity in Children, LOGIC study).[21] This programme of research utilises a

**Table 1** List of variables to be extracted for analysis for CYP aged 0–13 years who attended specialist gender services in the UK and the Netherlands between 2009 and 2017

| Variable | Data source | Level of data | Values |
|---|---|---|---|
| **Demographic** | | | |
| Age at referral | Medical record | Scale | Age in years and months |
| Age attended first appointment | Medical record | Scale | Age in years and months |
| Ethnicity* | Medical record | Nominal | 1: White<br>2: Mixed<br>3: Asian or Asian British<br>4: Black or Black British<br>5: Chinese or other<br>6: Prefer not to say |
| Registered sex assignation at birth | Medical record | Nominal | 1: Female<br>2: Male |
| Travel distance to gender identity clinic* | Medical record | Scale | Distance in kilometres |
| Family composition 1: Living situation | Medical record | Nominal | 1: CYP lives with both biological parents<br>2: CYP lives with one biological parent<br>3: Other |
| Family composition 2: LGBT parent | Medical record | Nominal | 1: No<br>2: Yes |
| Family composition 3: CYP adopted | Medical record | Nominal | 1: No<br>2: Yes |
| Primary caregiver age and relationship to CYP at referral | Medical record | Scale, Nominal | 1: Mother<br>2: Father<br>3: Stepmother<br>4: Stepfather<br>5: Adoptive mother<br>6: Adoptive father<br>7: Foster mother<br>8: Foster father<br>9: Foster parent (unspecified)<br>10: Aunt<br>11: Uncle<br>12: Grandparent |
| Sibling(s) age and sex* | Medical record | Scale, Nominal | 1: Female<br>2: Male<br>3: Other<br>4: No sibling |
| **Diagnostic** | | | |
| Gender dysphoria diagnosis | Medical record | Nominal | 1: No<br>2: Yes |
| **Gender Identity** | | | |
| Current gender identity* | Medical record; gender identity interview | Nominal | 1: Female<br>2: Male<br>3: Non-binary |
| Social transition prior to first appointment | Medical record | Nominal | 1: No<br>2: Yes<br>3: Partial |
| Social transition (at any time)* | Medical record | Nominal | 1: No<br>2: Yes<br>3: Partial |
| **Emotional and Behavioural Functioning** | | | |
| Total problems | Child behaviour checklist; | Ordinal | N/A |

| Table 1 Continued | | | |
|---|---|---|---|
| **Variable** | **Data source** | **Level of data** | **Values** |
| Internalising difficulties | Youth self report; | Ordinal | N/A |
| Externalising difficulties | Teacher report form | Ordinal | N/A |
| Autism | | | |
| Autistic traits | Social Responsiveness Scale | Ordinal | N/A |
| Referral information | | | |
| Date of referral | Medical record | Scale | N/A |
| Date of first and last appointments | Medical record | Scale | N/A |
| Total no. of appointments | Medical record | Scale | N/A |
| Date of discharge* | Medical record | Scale | N/A |
| Reason for discharge* | Medical record | Nominal | 1: Discharged against professional advice<br>2: Discharged on professional advice<br>3: Inappropriate Referral<br>4: Patient moved out of the area<br>5: Patient non-attendance<br>6: Patient requested discharge<br>7: Transferred from CAMHS to local Adult Mental Health Services<br>8: Transferred to other healthcare provider not medium/high secure<br>9: Patient not yet discharged |
| Healthcare resource use | | | |
| Type of clinical appointments | Medical record | Nominal | T&P NHSFT:<br>1: Early Liaison with endocrine clinic for <15 yrs[a]<br>2: Liaison with endocrine clinic for 15+ yrs[a]<br>3: GIDS Outreach Assessment[b]<br>4: GIDS Standard Assessment[b]<br>5: GIDS Treatment Outreach[c]<br>6: GIDS Treatment Standard[c]<br>7: GIDS Young Persons Group[d]<br>8: Group (not specified)[d]<br>9: GIDS Transitions Appointment[c]<br>10: Endocrinology (15+ yrs)[a]<br>11: Endocrinology (<15 years)[a]<br>12: Child, Young Adults and Families (CYAF) Assessment[b]<br>13: CYAF Individual therapy once per week[c]<br>14: CYAF Family Therapy[c]<br>AUMC:<br>1: Intake psychology[b]<br>2: Consultation psychology[b]<br>3: Psychological assessment[b]<br>4: Screening psychiatry[b]<br>5: Consultation psychiatry[b]<br>6: New patient endocrinology[a]<br>7: Consultation endocrinology[a]<br>8: Group consultation endocrinology[a]<br>9: DEXA scan[a]<br>10: Labs[a]<br>11: Consultation fertility[a] |
| Date of appointments/frequency | Medical record | Scale | N/A |
| Duration of appointments | Medical record | Scale | Time in minutes |

Continued

| Table 1 | Continued | | |
| --- | --- | --- | --- |
| Variable | Data source | Level of data | Values |
| Patient attendance | Medical record | Nominal | 1: Attended<br>2: Cancelled<br>3: Did not attend |
| Clinician attendance† | Medical record | Nominal | 1: Psychologist<br>2: Psychologist assistant<br>3: Psychiatrist<br>4: Endocrinologist<br>5: Doctors assistant<br>6: Nurse<br>7: Gynaecologist<br>8: Sexologist<br>9: Doctor<br>10: Unknown |

Appointment type groupings: [a]endocrinology/medical; [b]assessment; [c]psychosocial treatment; [d]group.
*Data available for T&P NHSFT only.
†Data available for AUMC only.
AUMC, Amsterdam University Medical Centre; CAMHS, Child and Adolescent Mental Health Services; CYP, children and young people; GIDS, gender identity development service; LGBT, lesbian, gay, bisexual, transgender; N/A, not available; NHSFT, National Health Service Foundation Trust; T&P, Tavistock & Portman.

mixed methods approach, incorporating both quantitative and qualitative longitudinal studies to investigate the experiences, outcomes and well-being of families referred to the UK GIDS.

## Study population
The study population will consist of all CYP aged ≤13 years who attended at least one appointment at a specialist gender service (GIDS or AUMC) between 2009 and 2017 and were recorded in the electronic patient records system used by the services. This will include approximately 1040 CYP from GIDS and 529 CYP from AUMC. CYP with differences in sex development and those referred to the service to obtain support with a parent undergoing gender transition will be excluded from the analyses.

## Data source
The study will consist of a retrospective analysis of routinely collected clinical data extracted from both the Tavistock and Portman NHSFT and the AUMC in the Netherlands between 2009 and 2017. The UK GIDS was established in 1989 and is currently one of the largest, if not the largest, specialist gender clinic for CYP in the world. It is a nationally commissioned service covering England, Wales, Northern Ireland and in part, Scotland and the Republic of Ireland, through a series of outreach clinics and main hubs in London and Leeds. The Center of Expertise on Gender Dysphoria in the Netherlands was established in 1988, and is one of the oldest and most established clinics. These two sites were chosen as they represent two of the largest and longest serving specialist clinics in Europe for CYP seeking support relating to their gender identity. These services are therefore uniquely placed to undertake a retrospective analysis of the characteristics and outcomes of CYP who attended the clinics

across an 8-year period. Completion of assessment and outcome measures such as the CBCL, YSR, TRF and SRS was entirely voluntary and not a condition of receiving care. The data from these measures were collected retrospectively for this study.

## Procedure
The research teams at each respective site (the Tavistock and Portman NHSFT in the UK and AUMC) will submit a request for data extraction to their local informatics team. Data will be extracted by each local informatics team via their electronic service user record software and entered into a comma-separated values (CSV) formatted dataset. Members of the research teams at each respective site will manually input any data into the datasets which cannot be extracted by the informatics teams, such as handwritten or typed information obtained from assessment reports. Data for social transition will be handsearched by the UK research team from medical records, although the content of which can vary enormously from patient-to-patient and by clinician. All identifiable data will be held in a password-protected database on an encrypted NHS server at the Tavistock and Portman NHSFT until the dataset has been finalised and is ready for analysis. Once the dataset is complete, all identifiable data (NHS patient IDs, dates of birth and all free-text responses) will be removed. The anonymised dataset will then be uploaded and stored onto an encrypted and General Data Protection Regulation (GDPR)-compliant data portal (Data Safe Haven) so that the statistical team at University College London (UCL) PRIMENT's Clinical Trials Unit can access the dataset for analysis. A data sharing agreement will be in place between the Tavistock and Portman NHSFT and the AUMC, each as data controllers. The sites will share ownership of the anonymised datasets once analysis is

complete and these datasets will be retained for no longer than 20 years. A collaboration agreement will also be enforced between all participating sites, identifying UCL as the data processor. The data processor shall destroy the data on request by the Tavistock and Portman NHSFT. The study will run for approximately 2 years (2019–2021).

## Analysis plan

Data will be analysed in STATA by the statistical team at UCL PRIMENT's Clinical Trials Unit. Characteristics of the CYP will be described using mean (SD), median (IQ range) or frequencies (proportion), as appropriate. In order to address study aim (2), the proportion of CYP who experience gender dysphoria, socially transition, access physical/medical treatment (ie, attend a paediatric endocrinology clinic) and have co-occurring autistic traits (as measured by the SRS) will be estimated, along with their 95% CIs. In order to address study aim (3), descriptive characteristics of the CYP who attend a paediatric endocrinology clinic will be compared with those for CYP who do not attend such a clinic. Regression models will be used to examine factors that are associated with referral to a paediatric endocrinology clinic. Factors which are likely to be explored within these models include: (1) family composition, (2) social transition, (3) emotional and behavioural functioning of CYP (measured by CBCL, YSR and TRF) and (4) autistic traits (measured by the SRS). Regression analyses will be adjusted for year of referral.

In relation to study aim (4), the average number of contacts with the service will be reported for each type of appointment. To establish healthcare resource costs, data pertaining to individual appointments (ie, assessment appointment; psychosocial treatment appointment; group appointment and endocrine appointment) from the time of referral until discharge from the service will be analysed. These will be costed based on information provided by the service on the number, profession and grade of clinical involvement for each appointment type and using information from the personal social service resource unit (PSSRU) to calculate cost per minute.[22] The cost per minute for each appointment type will then be multiplied by the duration of appointment, as recorded in patient files. We will conduct a sensitivity analysis using only service provided costs and only NHS Reference Costs.[22] The Netherlands will be costed based on PSSRU costing, with a sensitivity analysis using Netherlands-specific wages. Costed clinic appointments will then be summed together to calculate the total cost of care for each CYP and divided by contact time to adjust for patients with longer follow-ups. Average total costs of care and appointments will be reported for young people who attended a paediatric endocrinology clinic versus those who did not. These will be reported separately for the UK and the Netherlands. Further information regarding appointment types is provided in online supplemental appendix 1.

In order to address study aim (5), information such as (1) costs of care; (2) country of care; (3) care pathway (the type of treatment and/or support that the CYP receives throughout their time with the service); (4) outcomes including well-being (measured by CBCL, YSR and TRF); and (5) potential predictors of costs and outcomes (eg, age at first appointment, gender dysphoria diagnosis and autistic traits) will be used to explore differences in costs between CYP who attend a paediatric endocrinology clinic and those who do not. The purpose of this analysis is to calculate the cost of care for each CYP from the beginning to end of their time with GIDS, and then report if different CYP have different costs because (1) of the country they are in; (2) they followed the endocrine pathway (or didn't); (3) other clinical factors such as the prevalence of autistic traits. Further information regarding care pathways is described in online supplemental appendix 2.

All of the analyses will be presented individually for each service and also combined where possible. Where appropriate, analyses will be reported by year (2009–2017). If necessary, previously identified differences in baseline presentation of CYP referrals to the two services will be taken into account in the analyses, as well as other differences between the clinics such as the time (year) at which early physical interventions are offered. Potential bias due to missing data will be investigated by comparing the characteristics of CYP who have completed the reported outcome measures to those who have incomplete or no outcome data. Outcome measure data (ie, CBCL, TRF, YSR and SRS) will only be included in analyses when ≥70% of the cohort have completed the measures.

## Patient and public involvement statement

The LOGIC study was developed in collaboration with UK GIDS users. The research proposal was also discussed at a stakeholder event involving trans youth organisations. The LOGIC study has a patient and public involvement (PPI) group, composed of parents and CYP who are participating in our ongoing longitudinal cohort study. Findings and outputs will be discussed with the study PPI group.

## ETHICS AND DISSEMINATION

This study has been approved by the Health Research Authority and London—Hampstead Research Ethics Committee as application 19/LO/0181. The study findings will be published in peer-reviewed journals and presented at both conferences and stakeholder events.

**Author affiliations**
[1]Research & Development Unit, Tavistock and Portman NHS Foundation Trust, London, UK
[2]Research Department of Clinical, Educational and Health Psychology, University College London, London, UK
[3]Children, Young Adults and Families Directorate, Tavistock and Portman NHS Foundation Trust, London, UK
[4]Gender Identity Development Service, Tavistock and Portman NHS Foundation Trust, London, UK
[5]Department of Statistical Science, University College London, London, UK

[6]Research Department of Primary Care and Population Health, University College London, London, UK

[7]Division of Psychiatry, Faculty of Brain Sciences, London, UK

[8]Department of Paediatric and Adolescent Endocrinology, University College London Hospitals NHS Foundation Trust, London, UK

[9]Autism Research Centre, Department of Psychiatry, University of Cambridge, Cambridge, UK

[10]Center of Expertise on Gender Dysphoria, Vrije Universiteit Amsterdam, Amsterdam University Medical Center, location VUmc, Amsterdam, The Netherlands

[11]Department of Medical Psychology, Amsterdam University Medical Center, location VUmc, Amsterdam, The Netherlands

[12]Department of Child and Adolescent Psychiatry, Amsterdam University Medical Center, location VUmc, Amsterdam, The Netherlands

[13]Institute of Population Health Sciences, University of Liverpool, Liverpool, UK

**Acknowledgements** We would like to acknowledge the highly significant contribution made by Professor Michael King, University College London, who was a co-investigator on the NIHR LOGIC study. Professor King sadly passed away while the proofs of this manuscript were being prepared.

**Contributors** EK, PC, RO, RH, RS, GB, SB-C, NdG, TS, AdV, BY and MK contributed to the conception and design of the protocol. EK is the chief investigator of the LOGIC study. EK, CL, HS, VR and LS drafted the manuscript. RO and VV provided expertise on statistical analysis. RH provided expertise on health economic analysis. All authors drafted or critically revised the protocol and approved the final version of the manuscript.

**Funding** This work was supported by the National Institute for Health Research, Health Services and Delivery Research grant number 17/51/19.

**Competing interests** None declared.

**Patient consent for publication** Not applicable.

**Provenance and peer review** Not commissioned; externally peer reviewed.

**ORCID iDs**
Eilis Kennedy http://orcid.org/0000-0002-4162-4974
Veronica Ranieri http://orcid.org/0000-0003-0528-8640
Rachael Hunter http://orcid.org/0000-0002-7447-8934

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
