## [Reviewer comments · BMJ Open]

ARTICLE DETAILS

TITLE (PROVISIONAL)	Longitudinal Outcomes of Gender Identity in Children (LOGIC): study protocol for a retrospective analysis of the characteristics and outcomes of children referred to specialist gender services in the UK and the Netherla
AUTHORS	Kennedy, Eilis; Lane, Chloe; Stynes, Hannah; Ranieri, Veronica; Spinner, Lauren; Carmichael, Polly; Omar, Rumana; Vickerstaff, Victoria; Hunter, Rachael; Senior, Robert; Butler, Gary; Baron-Cohen, Simon; De Graaf, Nastasja; Steensma, Thomas; de Vries, Annelou; Young, Bridget; King, Michael

VERSION 1 – REVIEW

REVIEWER	Ashley, Florence University of Toronto
REVIEW RETURNED	20-Jul-2020

GENERAL COMMENTS	The study protocol is promising, and I recommend acceptance conditional upon revisions. The introduction does not profile a full picture of the existing knowledge base. While it is indeed 'limited', there are nonetheless many studies that have been conducted on the demographic and outcome data of trans children and youth from not only the teams involved in this study, but also others (Trans Youth CAN!, UWashington lab, various 'gender affirmative clinics' in the United States, etc.). At times, it read as though the authors were playing up the limited nature of the knowledge base to legitimate the study, the importance of which was not always thoroughly explained. The authors should make clearer why they chose these aims and methods for measuring them. Why is demographic and outcome data important to clinical pathways? What do they hope to achieve with the study? Do they seek differences in outcomes that would allow singling out subgroups as ineligible for transition-related care? Why is comparing the demographics of the clinic CYP to the general population demographics relevant, if overall trans populations may or may not share the demographics of the general population? Why are CBCL, YSR, and TRF adequate and well-adapted proxies or measures of wellbeing and/or functioning for gender non-conforming and trans youth? As it stands, the protocol leaves readers with too much guesswork as to why the study is being done. The protocol should include a broader discussion of limitations. Currently, limitations are only discussed in the 'article summary' and identify missing data as the main limitation (page 7). I can see at least a few more limitations regarding construct and external
--

validity (including the use of CBCL, YSR, and TRF as measures of wellbeing, which each have distinct limitations that should be acknowledged). The authors also don't seem to explain how missing data will be taken into account in the analyses.

I also include more specific comments to improve the protocol:

Page 8, line 22: The expression "assigned females at birth" should be replaced by "CYP assigned female at birth" in line with recommendations not to refer to trans boys and non-binary individuals as 'females'.

Page 8, lines 29-35: The authors should explain why it is critical to understand the characteristics and profiles of youth. Is it because the prior study doesn't sufficiently establish that there is no change in demographics or gender dysphoria? In what ways does the change in demographic or gender dysphoria matter among referrals matter?

Page 8, lines 41-42: The authors should adopt more neutral wording when referring to youth. For instance, "difficulties pertaining to gender identity" may imply a struggle with one's identity whereas many may instead be struggling with access to transition-related care and are not having any difficulties in relation to their gender identity per se.

Page 8, lines 44-56: The authors should describe why these assessment protocols are systematically undertaken and what role they play in the provision of care.

Page 9, lines 22-24: Since the source used does not provide a reference for its statement, is 4 years old, and mentions physical health and social functioning in the background section of the paper, I would recommend finding more recent and higher quality literature on the topic if possible. Some studies that came out after the source (e.g. 10.20319/pijss.2017.32.19701985) touch on social functioning.

Page 10, lines 3-8: It's not immediately clear how having demographics data and overall outcome information would help tailor care. Many studies have looked at the demographics and outcome data of trans people (notably adolescents and adults) but have failed to do much more than point out whether transition is 'on average' good or bad. What specifically does the study hope to find that may help tailor care? Does it hope to find subgroups for whom transition is on average harmful? Does it hope to confirm the general trend towards allowing social transition by showing that few youths are harmed by it? Etc.

Page 12, lines 3-6: Authors should include a reference to support the statement that the Dutch Protocol is widely adopted (verb tense suggests present).

Page 12, lines 43-58: Authors should specify what they consider to be identifiable data, how they account for the risk of re-identification given how small trans populations are, how the data processing agreement will be enforced and what the content of the agreement will include (in rough terms), and how long the data will be kept by the Tavistock & Portman NHSFT or UCL following completion of the study.

	Page 13, lines 33ff: Costs should be broken down based on source. For instance, costs could turn out to be much lower if less intense assessment procedures were adopted and may not reflect the actual health needs of the patients. Page 14, lines 25-26: The general nature of the stakeholders should be specified. Are they trans organisations, parents of trans youth organisations, general LGBT organisations? Page 14, lines 31-35: Given the study, it stands to reason that informed consent will not be asked of participants. The authors should briefly mention it. Page 14, lines 34-39: Will the findings be published in open access? Will the results be translated into an accessible language and format for stakeholders? If no, why not?
--	---

REVIEWER	Kuper, Laura University of Texas Southwestern Medical School, Psychiatry
REVIEW RETURNED	23-Jul-2020

GENERAL COMMENTS	I agree with the authors regarding the importance of better understanding the trajectories and associated service needs of children presenting for gender affirming care. The relatively large size of the clinics in UK and the Netherlands and the amount of data captured within the medical records are strengths of the current protocol. Adding further clarification and discussion in the following areas would further strengthen the protocol:  1. The rationale for calculation of costs associated with receiving clinic care is unclear and does not seem to fit with the focus of the other study aims. Further, why is the comparison of those who receive endocrine care (vs. other types of care) of particular interest? 2. Some variables would benefit from inclusion of a rationale: Family composition, parent/caregiver age, sibling sex and age, duration of appointments. 3. Some variables would benefit from further description: Types of clinical appointments, patient attendance, clinician attendance. In general, the timeline, content, and progression of appointments is unclear. 4. Further description of how social transition will be operationalized is particularly important. Social transition is complex, with a number of possible steps (e.g., changing name, pronoun, clothing, hairstyle) that may unfold differently and non-linearly across settings (e.g., may be using affirmed name/pronoun with friends and family but not at school). Attempts to condense into a single dichotomous variable is very concerning in light of this complexity (e.g., different aspects of social transition may be differentially related to other variables of interest; researchers will be imposing assumptions about what aspects of social transition should be considered most impactful; lack of clarity may lead readers to make a range of assumptions that may not be accurate). Identifying specific steps (e.g., change in pronouns) and settings (e.g., across all settings, with family only) would strengthen inclusion of these variables. For example, see:
--

- Kuper, L. E., Lindley, L., & Lopez, X. (2019). Exploring the gender development histories of children and adolescents presenting for gender affirming medical care. *Clinical Practice in Pediatric Psychology*, 7(3), 217.

How important aspects of social transition are to youth is also critical to understanding their trajectories (e.g., a youth who feels strongly about wanting to change their pronoun versus a youth who is not interested or a youth who is unsure about making this change). Similarly, levels of distress with sex characteristics and desires for medical treatment are also diverse and equally critical to understanding trajectories. Efforts should be made to attempt to capture these experiences with as much nuance as possible. When analyzing, interpreting, and discussing the results, the researchers should be particularly sensitive to the potential limitations of the data and careful to reflect on any assumptions they may be introducing.

5. Lack of inclusion of variables reflecting gender affirmation (e.g., parent and friend support, others use of correct name/pronoun), gender related minority stress (e.g., victimization, lack of accommodations), and resiliency factors (e.g., connection to transgender community) is a significant limitation given these variables are likely to have a major impact on youth's mental health and possibly their engagement in care. While including these variables is not possible within the scope of the current protocol, the impact of these variables still warrants discussion (particularly in future publications).

6. Some patients will have presented for care at the start of the data collection period (2009) while others will have just presented for care at the end of data collection (2017). These differences in timeline would impact any analyses related to number of appointments, discharge (date/reason), type of appointments, appointment frequency, attendance, and social transition (at any time). How will these differences be accounted for (e.g., in comparing those who did or did not receive care in the endocrinology clinic and in identifying factors related to service use)?

7. Concerns have been raised about past retrospective studies that have made assumptions about the reasons for lack of follow up in care. Most notably, that lack of follow up indicates that youth experienced a change in gender identity or dysphoria and no longer needed services when other explanations could be equally as plausible (e.g., lack of resources or support for continuing to engage in care; intensification of stigma or pressure to conform; discomfort with the clinic process; lack of nuance in identifying youth's experience of gender and how it may pertain to future treatment goals). How will the authors avoid making such problematic assumptions? Please consider:

- Julia Temple Newhook, Jake Pyne, Kelley Winters, Stephen Feder, Cindy Holmes, Jemma Tosh, Mari-Lynne Sinnott, Ally Jamieson & Sarah Pickett (2018) A critical commentary on follow-up studies and "desistance" theories about transgender and gender-nonconforming children, *International Journal of Transgenderism*, 19:2, 212-224, DOI: 10.1080/15532739.2018.1456390
- Kelley Winters, Julia Temple Newhook, Jake Pyne, Stephen Feder, Ally Jamieson, Cindy Holmes, Mari-Lynne Sinnott, Sarah Pickett & Jemma Tosh (2018) Learning to listen to trans and

	gender diverse children: A Response to Zucker (2018) and Steensma and Cohen-Kettenis (2018), International Journal of Transgenderism, 19:2, 246-250, DOI: 10.1080/15532739.2018.1471767 8. Ethical concerns have also been raised by the lack of community engagement in research involving transgender populations. The protocol mentions external stakeholders. Who were these and how were they involved? What efforts are being made to ensure community engagement throughout the remainder of the research process? Who is a “service user co-applicant”? Please consider additional ethical recommendations: - Benjamin William Vincent (2018) Studying trans: recommendations for ethical recruitment and collaboration with transgender participants in academic research, Psychology & Sexuality, 9:2, 102-116, DOI: 10.1080/19419899.2018.1434558
--	---

VERSION 1 – AUTHOR RESPONSE

Reviewers' Comments

Reviewer 1:

1. The introduction does not profile a full picture of the existing knowledge base. While it is indeed 'limited', there are nonetheless many studies that have been conducted on the demographic and outcome data of trans children and youth from not only the teams involved in this study, but also others (Trans Youth CAN!, UWashington lab, various 'gender affirmative clinics' in the United States, etc.). At times, it read as though the authors were playing up the limited nature of the knowledge base to legitimate the study, the importance of which was not always thoroughly explained.

Thank you for highlighting this. Much of the existing literature has focused on adolescents so less is currently known about pre-pubertal and early pubertal children who are referred to the services. We have now stated this more clearly in the introduction.

We are aware of a number of longitudinal cohort studies which have recently started and are now ongoing. We have referenced some of these ongoing studies in the introduction. Whilst we recognise that other studies have previously been published, we hope that this study, which includes a large cohort of children from two of the longest established and largest specialist services in Europe will contribute further to understanding, particularly in relation to younger children.

2. The authors should make clearer why they chose these aims and methods for measuring them. Why is demographic and outcome data important to clinical pathways?

The aims and methods were developed as part of an externally funded and peer-reviewed grant application. We consider that it is important to explore outcome data in order to ensure that services can plan accordingly for needs. It can also be helpful to examine demographic information in order to understand whether particular groups may face barriers and to explore changes over time (e.g. age at time of referral). As services have seen a significant increase in referrals over the past few years, it is important to understand more about the CYP who are referred to the services and the support and treatment that they are likely to access. It will be useful to identify the proportion of CYP who access medical treatment, the proportion with co-occurring autism and whether social transition is related to service use. This will be beneficial for service planning and ensuring that care pathways are resourced adequately.

3. What do they hope to achieve with the study? Do they seek differences in outcomes that would allow singling out subgroups as ineligible for transition-related care?

We hope to achieve a better understanding of the CYP and their families who are referred to the service, particularly in relation to younger CYP for whom less is currently known. The study aims to generate information and evidence that can be used to inform service planning. Singling out subgroups as ineligible for transition-related care is not an aim of the study and it is not anticipated that the findings from the study will be used for this purpose.

4. Why is comparing the demographics of the clinic CYP to the general population demographics relevant, if overall trans populations may or may not share the demographics of the general population?

The study will compare demographic information between services in the UK and the Netherlands. Demographic information within each service will be compared by year to explore changes over time. Demographic information for the CYP cohort will not be compared to general population demographics.

5. Why are CBCL, YSR, and TRF adequate and well-adapted proxies or measures of wellbeing and/or functioning for gender non-conforming and trans youth?

These are the measures that are routinely used by the two services in the UK and the Netherlands so unfortunately, these are the only measures available for this retrospective analysis. We recognise that these measures may have limitations and are including a broader range of measures in our ongoing longitudinal cohort study.

6. The protocol should include a broader discussion of limitations. Currently, limitations are only discussed in the 'article summary' and identify missing data as the main limitation (page 7). I can see at least a few more limitations regarding construct and external validity (including the use of CBCL, YSR, and TRF as measures of wellbeing, which each have distinct limitations that should be acknowledged).

As this is a study protocol, limitations have been outlined within the article summary (limited to five short bullet points) but a discussion section has not been included. We have included these measures as they are routinely used within the services. We will discuss these limitations when reporting and interpreting the findings of this study.

7. The authors also don't seem to explain how missing data will be taken into account in the analyses.

This has now been included (p.11).

8. Page 8, line 22: The expression "assigned females at birth" should be replaced by "CYP assigned female at birth" in line with recommendations not to refer to trans boys and non-binary individuals as 'females'.

Thank you, this has now been changed.

9. Page 8, lines 29-35: The authors should explain why it is critical to understand the characteristics and profiles of youth. Is it because the prior study doesn't sufficiently establish that there is no change in demographics or gender dysphoria? In what ways does the change in demographic or gender dysphoria matter among referrals matter?

Services have experienced a significant increase in referrals in recent years, particularly in relation to CYP assigned female at birth. It is important to establish the type of support that CYP are accessing so that services can plan accordingly. For example, whether there are differences in the type of support that younger and older CYP are likely to access. The study will also seek to identify the proportion of CYP referred to the services who receive a diagnosis of gender dysphoria.

10. Page 8, lines 41-42: The authors should adopt more neutral wording when referring to youth. For instance, “difficulties pertaining to gender identity” may imply a struggle with one’s identity whereas many may instead be struggling with access to transition-related care and are not having any difficulties in relation to their gender identity per se.

Thank you, this has been changed to “CYP seeking support relating to their gender identity.”

11. Page 8, lines 44-56: The authors should describe why these assessment protocols are systematically undertaken and what role they play in the provision of care.

The assessment protocols were developed to ensure that services are using clinically useful assessment measures and to enable comparisons between different services (Dèttore et al., 2015). Further detail has now been included.

12. Page 9, lines 22-24: Since the source used does not provide a reference for its statement, is 4 years old, and mentions physical health and social functioning in the background section of the paper, I would recommend finding more recent and higher quality literature on the topic if possible. Some studies that came out after the source (e.g. 10.20319/pijss.2017.32.19701985) touch on social functioning.

Thank you for this suggestion. More recent literature has now also been included.

13. Page 10, lines 3-8: It’s not immediately clear how having demographics data and overall outcome information would help tailor care. Many studies have looked at the demographics and outcome data of trans people (notably adolescents and adults) but have failed to do much more than point out whether transition is ‘on average’ good or bad. What specifically does the study hope to find that may help tailor care? Does it hope to find subgroups for whom transition is on average harmful? Does it hope to confirm the general trend towards allowing social transition by showing that few youths are harmed by it?

As noted, much of the existing literature relates to adolescents and adults so this study aims to address a gap in understanding by looking at younger children who are referred to the service. It is anticipated that the findings will provide insights into this younger cohort, specifically in relation to age at referral, gender dysphoria, social transition and co-occurring conditions (e.g. autism).

14. Page 12, lines 3-6: Authors should include a reference to support the statement that the Dutch Protocol is widely adopted (verb tense suggests present).

Thank you for highlighting this, this has now been amended.

15. Page 12, lines 43-58: Authors should specify what they consider to be identifiable data, how they account for the risk of re-identification given how small trans populations are, how the data processing agreement will be enforced and what the content of the agreement will include (in rough terms), and how long the data will be kept by the Tavistock & Portman NHSFT or UCL following completion of the study.

Further detail regarding the identifiable information to be removed (NHS patient IDs, dates of birth and all free-text responses), the terms of the data sharing and collaboration agreement, and the length of time data will be kept have now been added to pages 9 and 10.

We consider the risk of re-identification to be extremely low. The UK datasets containing identifiable patient information will be stored on a secure Tavistock & Portman NHSFT server in password-protected files. Only the immediate research team will be able to access these. The datasets, including VU Medical Centre's, will be fully anonymized by the respective research teams before being sent to the UCL PRIMENT statisticians for analysis via UCL's encrypted data portal. The combined datasets will contain >1500 participants from two clinics, making re-identification very unlikely without access to electronic patient record systems.

16. Page 13, lines 33ff: Costs should be broken down based on source. For instance, costs could turn out to be much lower if less intense assessment procedures were adopted and may not reflect the actual health needs of the patients.

There are three different appointment types at the Tavistock & Portman NHSFT: Standard appointment; Outreach appointment; and Group appointment. We have supplemented this information with NHS Reference costs to divide further into endocrinology costs and therapy costs. This will be tested using sensitivity analysis. Additional information has been added to the manuscript to clarify this (p. 10).

17. Page 14, lines 25-26: The general nature of the stakeholders should be specified. Are they trans organisations, parents of trans youth organisations, general LGBT organisations?

This study and the wider programme of research included in the LOGIC study were developed in collaboration with parents of young people attending the service and a trans service user co-applicant and co-author (TW). The research proposal was also discussed at a stakeholder event involving trans youth organisations. The study has been reviewed by the study steering committee.

18. Page 14, lines 31-35: Given the study, it stands to reason that informed consent will not be asked of participants. The authors should briefly mention it.

Informed consent is not required for this study as the data are routinely collected from the services and will be analysed for health or social care purposes; public health purposes; archiving in the public interest, scientific or historical research purposes or statistical purposes; and for reasons of substantial public interest.

19. Page 14, lines 34-39: Will the findings be published in open access? Will the results be translated into an accessible language and format for stakeholders? If no, why not?

We will endeavour where possible to publish the findings open access. We also plan to disseminate the findings in an accessible format for stakeholders (e.g. on our study website).

Reviewer 2:

1. The rationale for calculation of costs associated with receiving clinic care is unclear and does not seem to fit with the focus of the other study aims. Further, why is the comparison of those who receive endocrine care (vs. other types of care) of particular interest?

We anticipated that the cost of clinic care may differ between the UK and the Netherlands and therefore believed this was an aspect of care worth exploring. As there is considerable debate around young people accessing endocrine care and as access may differ between the UK and the Netherlands, we therefore considered that this was something worth focusing on.

2. Some variables would benefit from inclusion of a rationale: Family composition, parent/caregiver age, sibling sex and age, duration of appointments.

These variables have been included as they are routinely collected within the services.

3. Some variables would benefit from further description: Types of clinical appointments, patient attendance, clinician attendance. In general, the timeline, content, and progression of appointments is unclear.

Thank you for this suggestion. Further information has been added to Table 1 detailing how data for each variable will be coded.

4a. Further description of how social transition will be operationalized is particularly important. Social transition is complex, with a number of possible steps (e.g., changing name, pronoun, clothing, hairstyle) that may unfold differently and non-linearly across settings (e.g., may be using affirmed name/pronoun with friends and family but not at school). Attempts to condense into a single dichotomous variable is very concerning in light of this complexity (e.g., different aspects of social transition may be differentially related to other variables of interest; researchers will be imposing assumptions about what aspects of social transition should be considered most impactful; lack of clarity may lead readers to make a range of assumptions that may not be accurate). Identifying specific steps (e.g., change in pronouns) and settings (e.g., across all settings, with family only) would strengthen inclusion of these variables.

Thank you for highlighting this. We fully agree that social transition status is multi-faceted and that, ideally, the various elements of social transition (e.g., name, pronouns, and clothing) would be coded individually and separately for different contexts. As data for this variable will be hand-searched by the UK research team from medical records, content of which can vary enormously from patient-to-patient and by clinician, it will not be possible in this instance to code this variable in a more nuanced way. However, as part of the wider LOGIC project, our longitudinal cohort study is being conducted in parallel with this retrospective analysis study and will address these specific concerns.

Social transition will be recorded as yes, no or partial, and recorded for: 1. Social transition at time of referral; and 2. Social transition at any time during contact with the service (UK only) in order to capture some of the nuances described above. VUMC routinely record whether a CYP has socially transitioned at time of referral, but do not record social transition at any time during contact with their GID service. Further information regarding the coding of social transition for each clinic has been added to Table 1.

VUMC use the categories below to determine social transition status. To enable comparison between the two clinics, researchers in the UK team will code social transition information found in CYP medical records in this way. Several discussions have taken place between the two research teams to operationalise variables and ensure data is coded in the same way.

No = no indication of a social transition

Yes = name, pronoun, and appearance change to a gender other than that assigned at birth in ALL contexts (at home, school, community) - Including non-binary gender transitions

Partial = some indication of starting the process of social transition, but in limited contexts or only applied to certain gender expressions (e.g. appearance change, but no change of pronouns or name)

4b. How important aspects of social transition are to youth is also critical to understanding their trajectories (e.g., a youth who feels strongly about wanting to change their pronoun versus a youth who is not interested or a youth who is unsure about making this change). Similarly, levels of distress

with sex characteristics and desires for medical treatment are also diverse and equally critical to understanding trajectories. Efforts should be made to attempt to capture these experiences with as much nuance as possible. When analyzing, interpreting, and discussing the results, the researchers should be particularly sensitive to the potential limitations of the data and careful to reflect on any assumptions they may be introducing.

Thank you for highlighting this. As the study does not involve speaking directly with participants, unfortunately it was not possible to find out directly from CYP about social transition or their levels of distress. However, the limitations of this will be discussed when analysing, interpreting and discussing the results. As outlined above, this study is part of a wider programme of research investigating outcomes of gender identity in children and this information will be captured in the quantitative and qualitative cohort studies.

5. Lack of inclusion of variables reflecting gender affirmation (e.g., parent and friend support, others use of correct name/pronoun), gender related minority stress (e.g., victimization, lack of accommodations), and resiliency factors (e.g., connection to transgender community) is a significant limitation given these variables are likely to have a major impact on youth's mental health and possibly their engagement in care.

While including these variables is not possible within the scope of the current protocol, the impact of these variables still warrants discussion (particularly in future publications).

We fully agree that these are important variables to consider but are beyond the scope of this study. As suggested, the impact of these variables will be discussed when interpreting the findings of the study in future publications. These measures have been included as part of our ongoing longitudinal cohort study.

6. Some patients will have presented for care at the start of the data collection period (2009) while others will have just presented for care at the end of data collection (2017). These differences in timeline would impact any analyses related to number of appointments, discharge (date/reason), type of appointments, appointment frequency, attendance, and social transition (at any time). How will these differences be accounted for (e.g., in comparing those who did or did not receive care in the endocrinology clinic and in identifying factors related to service use)?

The descriptive statistics will be performed using the whole cohort and by year of referral. The regression analyses will be adjusted for the year of referral. Further information has now been included in the manuscript.

7. Concerns have been raised about past retrospective studies that have made assumptions about the reasons for lack of follow up in care. Most notably, that lack of follow up indicates that youth experienced a change in gender identity or dysphoria and no longer needed services when other explanations could be equally as plausible (e.g., lack of resources or support for continuing to engage in care; intensification of stigma or pressure to conform; discomfort with the clinic process; lack of nuance in identifying youth's experience of gender and how it may pertain to future treatment goals). How will the authors avoid making such problematic assumptions?

Thank you for this helpful point. The aim of the study is to identify the characteristics and outcomes of CYP referred to the services. As noted, there are a number of explanations for lack of follow up but unfortunately it is not possible to capture these as part of this study. However, when interpreting the findings from this study, we will be sensitive to this wide range of explanations.

8. Ethical concerns have also been raised by the lack of community engagement in research involving transgender populations. The protocol mentions external stakeholders. Who were these and how

were they involved? What efforts are being made to ensure community engagement throughout the remainder of the research process? Who is a “service user co-applicant”?

Thank you for highlighting this. We have now added more detail in the manuscript. This study and the wider programme of research included in the LOGIC study were developed in collaboration with parents of young people attending the service and a trans service user co-applicant and co-author (TW). The research proposal was also discussed at a stakeholder event involving trans youth organisations. The research team meet every few months with the study steering committee and the LOGIC study patient and public involvement group, which is comprised of parents and CYP who are participating in our longitudinal cohort study.

VERSION 2 – REVIEW

REVIEWER	Ashley, Florence University of Toronto
REVIEW RETURNED	13-Sep-2020

GENERAL COMMENTS	I am grateful for the authors’ revisions, many of which substantially improve the protocol. However, I consider that many of the concerns raised during the previous stage of peer review were not adequately addressed and invite further revisions along those lines. Studies of this scope require significant reflection and theoretical grounding that seems to be unfortunately missing at this stage. On query #2, (The authors should make clearer why they chose these aims and methods for measuring them. Why is demographic and outcome data important to clinical pathways?), the authors explain that the aims and methods were developed as part of a grant application and reiterate their importance but don’t offer compelling rationales about why they are important. For instance, they state “[i]t will be useful to identify the proportion of CYP who access medical treatment, the proportion with co-occurring autism and whether social transition is related to service use” but don’t offer detailed explanations why that is the case. Furthermore, those are also only some of what is measured and, as responses to the other reviewer highlight, in at least some cases they suggest that outcomes are measured simply because the data is available—a rationale that I do not find adequate. On query #3, (What do they hope to achieve with the study? Do they seek differences in outcomes that would allow singling out subgroups as ineligible for transition-related care?) the authors articulate goals around ‘gathering information’. I find the response too abstract and would welcome more thorough reflection on the social value of the information (see Danielle Wenner, “The Social Value of Knowledge and International Clinical Research” on a very similar concern in the international research context), especially given baseline ethical concerns surrounding chart reviews (which have waived consent requirements) and the dangers that the information will be (mis)used to deny or oppose care to subgroups like autistic trans youth. On query #4, (Why is comparing the demographics of the clinic CYP to the general population demographics relevant, if overall trans populations may or may not share the demographics of the
---

	general population?) the authors merely say they will be comparing the UK and Netherlands demographics but don't answer the question of why they do so. On query #11, (The authors should describe why these assessment protocols are systematically undertaken and what role they play in the provision of care.) I retain significant concerns about the theoretical rationale for routinely employing these assessment protocols. The Dettore paper does not give adequate explanations for why these assessment protocols are used and I am highly concerned by the prospect that trans youth find themselves pressured to complete extensive tests that can then be used in research with waived consent requirements even though using those tests with every service user regardless of individual context is of dubious clinical usefulness. If the authors cannot offer a compelling clinical rationale for using those tests with every service user, presumably as a condition of being able to access transition-related care, then the tests start to look less like clinical tools and more like research tools and the grounds for obtaining waived consent becomes ethically questionable. On query #14, (Authors should include a reference to support the statement that the Dutch Protocol is widely adopted (verb tense suggests present).) the revised passage retains ambiguity and the source used in support is of limited usefulness. It is a seminal text, but one that is 8 years old, written by the team who pioneered the approach, and isn't a study of how many teams use the protocol. By contrast, the claim of the Dutch protocol's pervasiveness seems to clash with some recent guidelines like the ANZPATH Standards of Care for Youth and the Statement on gender-affirmative approach to care from the pediatric endocrine society special interest group on transgender health, both of which favour the gender-affirmative model. The absence of quantitative points of reference for the claim of 'widely adopted' and the significant critiques of the model undermine the appeal to authority being made in the sentence. I would recommend simply deleting that sentence. On query #16, I welcome the changes made. I would further suggest that the analysis include a breakdown of the cost associated with making users systematically complete the CBCL, YSR, TRF, and SRS tests.
--	---

REVIEWER	Kuper, Laura University of Texas Southwestern Medical School, Psychiatry
REVIEW RETURNED	15-Oct-2020

GENERAL COMMENTS	The phrase "experiencing difficulties in relation to their gender identity" is used at the start of the introduction. Similar wording has been fixed in other areas, e.g., "seeking support" My initial comment that the authors do not provide rationale for including cost estimates has not been adequately addressed. The authors commented: "the cost of clinic care may differ between the UK and the Netherlands and therefore believed this was an aspect of care worth exploring." – if this finding holds, what are the implications, e.g., how will this information be used (to help make the UK program more cost efficient)?
---

	“As there is considerable debate around young people accessing endocrine care and as access may differ between the UK and the Netherlands, we therefore considered that this was something worth focusing on.” It is unclear how debates surrounding the clinical necessity of endocrine care and the impact of access to such care relate to the need for cost estimates of such care. Hopefully the authors are not suggesting that higher costs associated with endocrine care may be a reason to deny providing such care. In contrast, as pointed out by reviewer 1, if lengthy assessment visits appear to be quite high in cost, this could be a justification for condensing these visits, particularly given the concerns that are increasingly being raised about the lack of clinical utility for aspects of these assessments (e.g., IQ testing). In addition to responding to these questions within the response to reviewers, incorporating them into the manuscript itself would provide important context to readers. The authors acknowledged that important variables relevant to the profiles and care pathways of youth are not included in the current protocol (e.g., social support, victimization, other mental health concerns such as suicide ideation & attempt, and more nuanced information regarding desired social transition steps and their progression). This would be important to note as a limitation. The authors make several references to this protocol being a part of the larger LOGIC study. More information about the LOGIC study, namely how the current protocol fits within this larger study, would provide helpful additional context. The introduction would benefit from providing a brief rationale for including a measure of autism spectrum traits. I want to make sure the authors are aware of a recent study that suggests that rates of autism spectrum traits identified by the SRS may be artificially inflated due to overlap with other emotional and behavioral difficulties: Leef, J. H., Brian, J., VanderLaan, D. P., Wood, H., Scott, K., Lai, M. C., ... & Zucker, K. J. (2019). Traits of autism spectrum disorder in school-aged children with gender dysphoria: A comparison to clinical controls. Clinical Practice in Pediatric Psychology, 7(4), 383. The authors state: “In relation to study aim (4), regression models will be used to examine factors that predict service use (i.e. the number and type of contacts with services).” However, the authors reported 14 to 17 different appointment types (depending on clinic) as well as variables reflecting number, length, and frequency of appointments, cancellations, and type of clinician in attendance. Can the authors provide more detail on the specific service use variables that will be examined (e.g., will certain appointment types be collapsed or excluded from the analyses)? The authors provide more information about how social transition was coded in the response to reviewers. It would be helpful to also include this within the manuscript, including the authors’ note that: “data for this variable will be hand-searched by the UK research team from medical records, content of which can vary enormously from patient-to-patient and by clinician.”
--	--

	I appreciate the authors' updates to the tables to reflect the type and values for each variable. However, the content of appointment types is still unclear, but critical to understanding the types of services that these clinics provide. For example, what is the difference between an outreach assessment, standard assessment, treatment outreach, treatment standard? The acronym CYAF is used but not defined. An additional chart providing brief descriptions of these appointment types would be helpful. Additionally, it is unclear how patients progress through appointment types. When presenting for care, is there a type of visit that everyone participates in? Are there different types of initial visits depending on how patients present to care and/or what they identify as their needs when presenting for care? How is the choice for subsequent appointments made? This information is important given the authors goal of examining care pathways and outcomes. Is information about Tanner stage collected? It is important to note that within a child sample, many may not have started puberty, in which case no endocrine treatment is required. This is a different scenario from a child who is actively in puberty but not desiring endocrine treatment. It seems like both of these groups will be collapsed when predicting use of endocrine services?
--	--

VERSION 2 – AUTHOR RESPONSE

Reviewers' Comments

Reviewer 1:

On query #2, (The authors should make clearer why they chose these aims and methods for measuring them. Why is demographic and outcome data important to clinical pathways?), the authors explain that the aims and methods were developed as part of a grant application and reiterate their importance but don't offer compelling rationales about why they are important. For instance, they state "[i]t will be useful to identify the proportion of CYP who access medical treatment, the proportion with co-occurring autism and whether social transition is related to service use" but don't offer detailed explanations why that is the case. Furthermore, those are also only some of what is measured and, as responses to the other reviewer highlight, in at least some cases they suggest that outcomes are measured simply because the data is available—a rationale that I do not find adequate.

The purpose of the study is to gain further insight into characteristics, over time, of CYP aged ≤13 years when referred to two of the world's largest gender identity development services, and to examine whether there are time trends in demographic, diagnostic, and treatment characteristics between 2009 and 2017. As noted in the manuscript, there is evidence that the demographics of young people referred to services worldwide is changing, both in age at referral and assigned gender at birth. In addition, many more younger children have made a full social transition before they are referred to services. The outcomes in relation to gender identification of young people attending specialist gender services can vary. In some cases a diverse gender identification to that assigned at

birth continues and in others it discontinues. It is therefore important to investigate whether there is any relationship between changing demographic and social factors and the outcome of gender identification over time.

The aims and methods were peer-reviewed as part of an externally funded grant application and we were awarded competitive funding for this study. Unfortunately, as these aims and methods have been agreed with our funder, we cannot deviate too much from the study protocol that has had both prior agreement from the funder and ethical approval. We completely agree that the study will be limited because of limitations in the available data collected by both services and the retrospective study design. It is for this reason we are undertaking a prospective longitudinal study, with a broader range of what we hope are meaningful outcome assessments and a companion longitudinal qualitative study where we learn directly from children, young people and their families about their experiences.

On query #3, (What do they hope to achieve with the study? Do they seek differences in outcomes that would allow singling out subgroups as ineligible for transition-related care?) the authors articulate goals around 'gathering information'. I find the response too abstract and would welcome more thorough reflection on the social value of the information (see Danielle Wenner, "The Social Value of Knowledge and International Clinical Research" on a very similar concern in the international research context), especially given baseline ethical concerns surrounding chart reviews (which have waived consent requirements) and the dangers that the information will be (mis)used to deny or oppose care to subgroups like autistic trans youth.

We hope that the study will be informative regarding the characteristics, over time, of CYP aged ≤ 13 years when referred to two of the world's largest gender identity development services. The study will in no way seek differences in outcomes that would allow singling out of particular subgroups as ineligible for transition-related care. The study has undergone ethical review by an ethics committee and has obtained ethical approval. Thank you very much for drawing our attention to the Danielle Wenner paper. This paper helpfully raises a number of important ethical issues regarding the potential for information from research studies to be misused in ways that are not beneficial to important community stakeholders. We agree it is important to be aware of and sensitive to these concerns as potential issues for our study. We have therefore included community engagement and input from stakeholders in the design of our study. Our research team includes a peer researcher and in addition, we have an active and engaged study patient and public involvement (PPI) group, involving service users which meet on a regular basis and in addition service user representation on the study steering committee. We will work collaboratively with this PPI group to ensure that outputs and the interpretation of findings are meaningful and of value to the children, young people and families who are the focus of our study.

On query #4, (Why is comparing the demographics of the clinic CYP to the general population demographics relevant, if overall trans populations may or may not share the demographics of the general population?) the authors merely say they will be comparing the UK and Netherlands demographics but don't answer the question of why they do so.

As described within the manuscript, the services in the UK and the Netherlands are two of the longest established and largest specialist services in Europe for CYP seeking support in relation to their gender identity. Both of the services follow a similar assessment protocol and both have consistently used the same outcome measures over the past 8 years, enabling comparisons to be made between the services. Comparing these services will provide insights into certain demographic differences that may be specific to each clinic (e.g. age of referral). This demographic information (e.g. around number of children of a particular age referred to each clinic) is important information in relation to planning and resourcing services and has implications for the services CYP and families receive (e.g. recent increases in number of referrals has led to lengthy waiting times for CYP). Furthermore, this information will be helpful in identifying potential differences or impediments in access to specialist services for particular groups (e.g. by age or other factors).

On query #11, (The authors should describe why these assessment protocols are systematically undertaken and what role they play in the provision of care.) I retain significant concerns about the theoretical rationale for routinely employing these assessment protocols. The Dettore paper does not give adequate explanations for why these assessment protocols are used and I am highly concerned by the prospect that trans youth find themselves pressured to complete extensive tests that can then be used in research with waived consent requirements even though using those tests with every service user regardless of individual context is of dubious clinical usefulness. If the authors cannot offer a compelling clinical rationale for using those tests with every service user, presumably as a condition of being able to access transition-related care, then the tests start to look less like clinical tools and more like research tools and the grounds for obtaining waived consent becomes ethically questionable.

Thank you for highlighting this. We understand that there may be concerns around the theoretical basis for these assessment protocols but this is unfortunately beyond the remit of this study and we are constrained by the assessment protocols that are currently agreed and used by the services. However, this is a helpful point to consider and we acknowledge that the assessment protocols may change in the future. We also acknowledge that 'assessment' is potentially an unhelpful term as it can have different meanings for different services. We appreciate the concerns raised in relation to CYP feeling pressured to complete extensive tests but completion of these measures is entirely voluntary and there is no requirement for CYP to complete measures as a condition of receiving care. We therefore anticipate that there will be variability in the number of CYP who will complete these assessments. We will discuss the limitations of these measures and the completion rates when discussing the findings from the study.

On query #14, (Authors should include a reference to support the statement that the Dutch Protocol is widely adopted (verb tense suggests present).) the revised passage retains ambiguity and the source used in support is of limited usefulness. It is a seminal text, but one that is 8 years old, written by the team who pioneered the approach, and isn't a study of how many teams use the protocol. By contrast, the claim of the Dutch protocol's pervasiveness seems to clash with some recent guidelines like the ANZPATH Standards of Care for Youth and the Statement on gender-affirmative approach to care from the pediatric endocrine society special interest group on transgender health, both of which favour the gender-affirmative model. The absence of quantitative points of reference for the claim of 'widely adopted' and the significant critiques of the model undermine the appeal to authority being made in the sentence. I would recommend simply deleting that sentence.

Thank you, this sentence has been deleted.

On query #16, I welcome the changes made. I would further suggest that the analysis include a breakdown of the cost associated with making users systematically complete the CBCL, YSR, TRF, and SRS tests.

It is unfortunately not possible to calculate the specific costs associated with completion of these outcomes measures because completion of these measures is entirely voluntary and measures are not completed systematically by all service users. We will only be able to obtain costs affiliated with different appointment types and the staff in attendance at those appointments: the services do not record at which appointment type the outcome measures were completed, and some measures, such as the TRF, are completed outside of the clinic by a teacher. However, these costs would be included as part of the total appointment time as they form part of standard care.

Reviewer 2:

The phrase "experiencing difficulties in relation to their gender identity" is used at the start of the introduction. Similar wording has been fixed in other areas, e.g., "seeking support"

Thank you, this has now been amended accordingly.

My initial comment that the authors do not provide rationale for including cost estimates has not been adequately addressed. The authors commented:

"the cost of clinic care may differ between the UK and the Netherlands and therefore believed this was an aspect of care worth exploring." – if this finding holds, what are the implications, e.g., how will this information be used (to help make the UK program more cost efficient)?

“As there is considerable debate around young people accessing endocrine care and as access may differ between the UK and the Netherlands, we therefore considered that this was something worth focusing on.” It is unclear how debates surrounding the clinical necessity of endocrine care and the impact of access to such care relate to the need for cost estimates of such care. Hopefully the authors are not suggesting that higher costs associated with endocrine care may be a reason to deny providing such care. In contrast, as pointed out by reviewer 1, if lengthy assessment visits appear to be quite high in cost, this could be a justification for condensing these visits, particularly given the concerns that are increasingly being raised about the lack of clinical utility for aspects of these assessments (e.g., IQ testing). In addition to responding to these questions within the response to reviewers, incorporating them into the manuscript itself would provide important context to readers.

Thank you for raising these important points. Firstly we would like to confirm that we are in no way implying that the possibility of higher costs being associated with endocrine care would be a reason to deny providing such care. Indeed it may well be that access to endocrine care reduces costs overall because of health benefits elsewhere. With regard to the specific question of why we are focusing on health care costs in the UK and the Netherlands, this is in part because the commissioning brief for the research grant we were awarded requested that this was something that would be included as an aspect of our research. The intention behind this is to explore how the organisation and resourcing of healthcare might improve outcomes for young people. This research is led by a research team independent of those involved in service delivery and our primary focus is to undertake research that will be of benefit to young people and their families referred to services. We can clarify that were it determined that ‘lengthy assessment visits’ ‘appear to be at quite high cost’ this would certainly **not** ‘be used as a justification for condensing these visits’. The purpose of our research is to generate knowledge that will be of direct benefit to CYP accessing services. In relation to the issue raised regarding the clinical utility of assessments, IQ is not assessed by the service in the UK and this is not something we will be looking at. We are aware of the potential limitations of some of the assessments undertaken but as this is a retrospective study this is not something we are in a position to influence. However it is something we will endeavour to address in the prospective longitudinal study we are undertaking.

The authors acknowledged that important variables relevant to the profiles and care pathways of youth are not included in the current protocol (e.g., social support, victimization, other mental health concerns such as suicide ideation & attempt, and more nuanced information regarding desired social transition steps and their progression). This would be important to note as a limitation.

Thank you, this has now been included as a limitation.

The authors make several references to this protocol being a part of the larger LOGIC study. More information about the LOGIC study, namely how the current protocol fits within this larger study, would provide helpful additional context.

Thank you, additional information has now been included about the wider LOGIC study to provide additional context.

The introduction would benefit from providing a brief rationale for including a measure of autism spectrum traits. I want to make sure the authors are aware of a recent study that suggests that rates of autism spectrum traits identified by the SRS may be artificially inflated due to overlap with other emotional and behavioral difficulties:

Leef, J. H., Brian, J., VanderLaan, D. P., Wood, H., Scott, K., Lai, M. C., ... & Zucker, K. J. (2019). Traits of autism spectrum disorder in school-aged children with gender dysphoria: A comparison to clinical controls. *Clinical Practice in Pediatric Psychology*, 7(4), 383.

Thank you for this helpful suggestion. A brief rationale regarding the inclusion of an autism measure has now been added to the introduction. Limitations associated with this measure will be discussed when interpreting the findings, including reference to the suggested paper.

The authors state: “In relation to study aim (4), regression models will be used to examine factors that predict service use (i.e. the number and type of contacts with services).” However, the authors reported 14 to 17 different appointment types (depending on clinic) as well as variables reflecting number, length, and frequency of appointments, cancellations, and type of clinician in attendance. Can the authors provide more detail on the specific service use variables that will be examined (e.g., will certain appointment types be collapsed or excluded from the analyses)?

The average number of contacts with the service will be reported for each type of appointment, and, where possible, we will also examine the other variables listed above. This has now been clarified in the manuscript, and we hope the planned analysis in relation to each study aim is now clearer. To establish healthcare resource costs, data pertaining to individual appointments (i.e. standard appointment; outreach appointment; group appointment; endocrinology and therapy) from the time of referral until discharge from the service will be analysed. These will be costed based on information provided by the service on the number, profession and grade of clinical involvement for each appointment type and using information from the Personal Social Service Resource Unit (PSSRU) to calculate the cost per minute. The cost per minute for each appointment type will be multiplied by the duration of appointment as recorded in patient files. We will conduct a sensitivity analysis using only service provided costs and only NHS Reference Costs. Netherlands will be costed based on PSSRU costing, with a sensitivity analysis using Netherlands specific wages. Costed clinic appointments will then be summed together to calculate the total cost of care for each CYP and divided by contact time

to adjust for patients with longer follow-ups. This information has been added to the manuscript (page 11).

The authors provide more information about how social transition was coded in the response to reviewers. It would be helpful to also include this within the manuscript, including the authors' note that: "data for this variable will be hand-searched by the UK research team from medical records, content of which can vary enormously from patient-to-patient and by clinician."

Thank you, we have now included further information within the manuscript.

I appreciate the authors' updates to the tables to reflect the type and values for each variable. However, the content of appointment types is still unclear, but critical to understanding the types of services that these clinics provide. For example, what is the difference between an outreach assessment, standard assessment, treatment outreach, treatment standard? The acronym CYAF is used but not defined. An additional chart providing brief descriptions of these appointment types would be helpful. Additionally, it is unclear how patients progress through appointment types. When presenting for care, is there a type of visit that everyone participates in? Are there different types of initial visits depending on how patients present to care and/or what they identify as their needs when presenting for care? How is the choice for subsequent appointments made? This information is important given the authors goal of examining care pathways and outcomes.

In the UK, following the NHS England service specification, all patients attend the service for assessment in the first instance. According to the service specifications, this typically consists of 3-6 appointments, with the number of appointments based on individual need, agreed with the young person and their family. At the end of the assessment, if the young person remains in contact with the service they either continue to explore their gender identity and options around this, or they may be referred to the endocrine clinic with ongoing exploration of gender and pathways and psychosocial support. The current NHS service specification can be found here: <https://www.england.nhs.uk/wp-content/uploads/2017/04/gender-development-service-children-adolescents.pdf>. Please note that this is currently undergoing a scheduled review.

As GIDS is a national service, 'outreach' assessment simply refers to the assessment being undertaken remotely from the main clinic base in London. After an assessment, if the CYP remains in contact with the service, appointment type then changes to 'treatment': 'treatment outreach' would indicate the treatment took place outside the main London clinic base, whereas 'treatment standard' would mean that the treatment took place at the London clinic. We will certainly provide additional descriptions when reporting findings. Thank you for highlighting that the acronym CYAF is not defined, this has now been done.

In the Netherlands, the first consultation ('intake') is always with a child and adolescent psychiatrist. The following appointments during the assessment phase are with a psychologist. This usually consists of 3-6 appointments (on a monthly basis), with the number of appointments based on individual need, and agreed with the young person and their family. At the end of the assessment, the majority of CYP are referred to endocrinology and also continue to see a psychologist for psychosocial support. The young person and their family are seen by the endocrinologist and psychologist every 3 months.

Is information about tanner stage collected? It is important to note that within a child sample, many may not have started puberty, in which case no endocrine treatment is required. This is a different scenario from a child who is actively in puberty but not desiring endocrine treatment. It seems like both of these groups will be collapsed when predicting use of endocrine services?

Information regarding tanner stage is not collected in this study. We agree this is a limitation and this will be acknowledged and discussed in interpreting the findings. This is something that will be looked at systematically in our prospective longitudinal study.

VERSION 3 – REVIEW

REVIEWER	Ashley, Florence University of Toronto
REVIEW RETURNED	15-Nov-2020

GENERAL COMMENTS	I appreciate the authors' revisions and responses to the queries. I retain two significant concerns. The authors point out that the tests used (CBCL, YSR, and TRF) are voluntary and are not a condition for receiving care. I appreciate their specification. However, I cannot in good conscience recommend publication unless the authors clearly explain the clinical rationale for the measures or seek individual consent for study participation. Waiver of informed consent for chart reviews is predicated on the assumption that information contained in charts are obtained for bona fide clinical purposes. If they are not gathered for bona fide clinical purposes, then participants' informed consent to have these measures taken and used in the context of research despite the absence of bona fide clinical purposes must be sought. (This is in relation to query #11) I also appreciate that the authors' aims and methods are constrained by funding approval and cannot be deviated from. However, they should still be able to clearly articulate what is the theoretical rationale behind these aims and methods. To give but one example, they state: "[i]t will be useful to identify the proportion of CYP who access medical treatment, the proportion with co-occurring autism and whether social transition is related to
---

	service use.” The authors should explain *why* it will be useful. Will it be useful to inform clinics’ distribution of resources between support for medical transition versus other forms of support alone? Is the goal of inquiring into the proportion of co-occurring autism to ascertain the need for additional training and resources specific to autistic CYP? Or do they expect that these answers may inform changes to their clinical approach (and if yes, on what grounds)? While the aims and methods cannot be changed, they can be explained. (This is in relation to query #1)
--	--

REVIEWER	Kuper, Laura University of Texas Southwestern Medical School, Psychiatry
REVIEW RETURNED	13-Jan-2021

GENERAL COMMENTS	I appreciate the ability to review this revised manuscript and I continue to agree that the information collected within the current protocol is of particular importance to the field. However, I am concerned that the authors only provided cursory responses to a number of areas of improvement that have been identified during the review process. I understand that the authors are somewhat constrained by the aims that they identified in a grant proposal; however, this should not preclude revisions to better explain their services, the rationale for their proposed analyses, and consideration of additional analyses that would be possible within the constraints of their dataset. From my perspective, such edits will strengthen the reach of the protocol to a broader range of interested parties that may not be as familiar with the specific organization of the clinics and the details of the types of care that they provide. While the authors provided some explanation about the care pathways present at each clinic within their response to reviewers, this information was not integrated into the manuscript. Given care pathways vary significantly across pediatric gender clinics and the goal of the protocol is to better understand the profile of CYP progressing through different care pathways, clear description of the care pathways is essential. This includes how families initially make contact with the clinic, what initial visit(s) are scheduled, what subsequent visit(s) are available (including visit types, frequency, and timing/relationship between visits -e.g., are certain visits required, do they occur in a specific progression), the general focus/goal(s) of each visit type, and how recommendations/referrals are made for each visit type. It may be helpful to group appointment types together to make it clearer, i.e., (a) initial/intake appointments, (b) assessment appointments, (b) optional other mental health appointments (consultation, therapy), (c) endocrine care, (d) optional other medical appointments. For VUMC, there are a range of medical appointment types that are unclear, particularly “reading status,” “clinical genetics,” “intercollegial consultation,” “emergency consultation,” “group consultation endocrinology.” It is also not clear why endocrinology is broken into under and over 15 (and in one clinic only) and how an outreach/standard assessment differs from a CYAF assessment. While the authors focus on identifying characteristics associated with attending endocrine appointments, identifying characteristics associated with attending mental health services could be equally beneficial to improving these services. If sample sizes permit, it would also be very helpful to know if there are any differences
---

	between those that remain engaged in care and those that stop attending or are “discharged against professional advice.” Explaining how the latter category is defined would be helpful. Beyond noting that the aim was included in a grant proposal, I still do not see any specific justification for including calculations surrounding the cost of care or information about how this data will be used to improve care. Why were costs associated with care identified as an important research question and why was endocrine care selected in particular (versus all forms of care or mental health services)? Once this data is collected, who will the authors be sharing it with and what are the potential goals and outcomes of sharing the data, including how might they improve care? Given readers interested in better understanding patient demographics and care pathways are likely to be at least somewhat different than those interested in specific cost estimates, it may be advisable to remove cost related aims from the present manuscript. This would also allow room to better specify care pathways and how clinical information will be coded and analyzed. Regarding study aim 5: (a) The proposed analyses do not match up clearly with the aim. The aim is focused on identifying the cost of care and cost of care is included as one variable potentially predicting who attended an endocrinology visit, but other variables are also included as predictors of attendance. (b) The authors state that they will include “outcomes including wellbeing (measured by CBCL, YSR and TRF)” as a predictor of attendance. When and how is this longitudinal data collected? (c) “Care pathway” and “potential predictors of costs and outcomes” are not defined. What is the relationship between the Adolescent Gender Identity Research group (AGIR) and the Logic study? The authors mention that the clinics are participants in both. I am curious about how many service users and organizations provided input into the development of the protocol and whether this included CYP or only parents. Was feedback provided at one point in time or on an ongoing basis? Can any of the data being collected within the protocol speak to the potential impact of the Bell vs Tavistock ruling, which severely limits access to gender affirming medical treatment prior to age 16? This would be of greatest importance and is particularly urgent given the many CYP whose medically necessary treatment is being halted or blocked.
--	--

VERSION 3 – AUTHOR RESPONSE

Reviewer: 1

Dr. Florence Ashley, University of Toronto

Comments to the Author:

I appreciate the authors' revisions and responses to the queries.

Thank you for noting this.

I retain two significant concerns. The authors point out that the tests used (CBCL, YSR, and TRF) are voluntary and are not a condition for receiving care. I appreciate their specification. However, I cannot in good conscience recommend publication unless the authors clearly explain the clinical rationale for the measures or seek individual consent for study participation. Waiver of informed consent for chart reviews is predicated on the assumption that information contained in charts are obtained for bona fide clinical purposes. If they are not gathered for bona fide clinical purposes, then participants' informed consent to have these measures taken and used in the context of research despite the absence of bona fide clinical purposes must be sought. (This is in relation to query #11).

We would like to thank the reviewer for highlighting these important issues and welcome the opportunity to provide further clarification.

With regard to the clinical rationale for the questionnaires used, the Child Behaviour Checklist (CBCL), along with the Teacher's Report Form (TRF), and the Youth Self-Report (YSR) forms are components of the Achenbach System of Empirically Based Assessment (ASEBA) widely used in clinical services in the UK, Europe and North America. We can confirm, as requested, that these questionnaire measures were administered by specialist gender services for clinical purposes. Scores on these questionnaires would have indicated to the clinical care team any behavioural, emotional, or social difficulties the CYP may have been experiencing, which could be important when planning the care and psychosocial support to be put in place. We can therefore provide assurance that these measures were administered for bona fide clinical purposes and these data were collected retrospectively for this study. As highlighted previously, completion of the measures is entirely voluntary and not a condition of receiving care. Thank you for drawing our attention to this we have now added further information to the manuscript make this clearer. (pg.9).

"Completion of assessment and outcome measures such as the CBCL, YSR, TRF, and SRS and was entirely voluntary and not a condition of receiving care. The data from these measures were collected retrospectively for this study."

In order to ensure that outcome measures are representative of the cohort and therefore provide meaningful information, scores on these measures will only be included in the analyses if completion rates are $\geq 70\%$. This is now stated in the manuscript (pg. 12).

“Outcome measure data (i.e. CBCL, TRF, YSR, and SRS) will only be included in analyses when $\geq 70\%$ of the cohort have completed the measures.”

The study was approved by the Research Ethics Committee (REC) and the Health Research Authority and London – Hampstead Research Ethics Committee (application 19/LO/0181). In line with UK guidance, de-identified patient information can be used for research, for the purpose of improving patient care or if it is in the public interest.

I also appreciate that the authors’ aims and methods are constrained by funding approval and cannot be deviated from. However, they should still be able to clearly articulate what is the theoretical rationale behind these aims and methods. To give but one example, they state: “[i]t will be useful to identify the proportion of CYP who access medical treatment, the proportion with co-occurring autism and whether social transition is related to service use.” The authors should explain **why it will be useful. Will it be useful to inform clinics’ distribution of resources between support for medical transition versus other forms of support alone? Is the goal of inquiring into the proportion of co-occurring autism to ascertain the need for additional training and resources specific to autistic CYP? Or do they expect that these answers may inform changes to their clinical approach (and if yes, on what grounds)? While the aims and methods cannot be changed, they can be explained. (This is in relation to query #1)**

We would like to thank the reviewer for raising these important questions. With regard to the theoretical rationale for our aims and method, our primary purpose is to describe the characteristics of children and young people attending specialist gender services in the UK and the Netherlands. The significant increase in referrals to gender services for children and young people both in the UK and internationally in recent years has led to questions around the proportion of CYP accessing medical treatment (puberty blockers and cross sex hormones), the proportion of CYP with co-occurring autism and the proportion who socially transition. These issues have been the subject of particular debate and controversy in the UK. The UK service has been the subject of a judicial

review: www.judiciary.uk/wp-content/uploads/2020/12/Bell-v-Tavistock-Clinic-and- (1 December 2020) and the health service in England have established an independent expert group chaired by Dr Hilary Cass to review practice in the field. We are also very aware of the impact the controversy has had internationally <https://epath.eu/joint-statement-regarding-medical-affirming-treatment-including-puberty-blockers-for-transgender-adolescents/>.

Our objectives are to better understand the characteristics of service users in order to help in the planning and organisation of services and to address the need for tailored support when required. This has now been stated clearly in the manuscript (pg. 7).

“It is anticipated that the proposed research will improve understanding of the characteristics of service users in order to help in the planning and organisation of services and to address the need for tailored support when required.”

This is in line with the specific NIHR programme which funds this research - HS&DR (Health Services and Delivery Research) - which focuses on producing evidence “to improve the quality, accessibility and organisation of health and social care services.” (<https://www.nihr.ac.uk/explore-nihr/funding-programmes/health-services-and-delivery-research.htm>).

We are mindful of the complexities around this and the potential dangers of over medicalising issues in an unhelpful way. This is why we are committed to ensuring that the voices and needs of children and young people with direct lived experience inform how study findings are used and applied. Our research programme includes a longitudinal qualitative study, led by the University of Liverpool, which involves children and young people aged 5-18 which will gather important data on the perspectives of children and young people themselves and this will also help in informing our approach.

We want to ensure that young people and their families using the service are fully engaged and involved in discussions around how the study findings could best be applied in a service context. For example, with regard to the question of resources for medical transition versus other forms of support, we will share findings with our study PPI group and will be guided by their views on this. Similarly with regard to the issue of co-occurring autism we plan to share our findings on this with young people themselves and the wider autism community to seek guidance regarding what specific support or resources might potentially be helpful.

It is also important to note that the LOGIC study programme of research is externally funded and involves a collaboration between the NHS, University College London and the Universities of Liverpool and Cambridge. Whilst the research team does include two clinicians (Dr Polly Carmichael and Professor Gary Butler) who work clinically within GIDS, the overwhelming majority of the co-investigator team are independent of the service, including an academic with lived experience (based at University College London). We hope that the independence of the research team alongside the active involvement of those with lived experience of services will ensure that the proposed research will best serve the children and young people referred to specialist gender services.

Reviewer:2

Dr. Laura Kuper, University of Texas Southwestern Medical School, Children's Health System of Texas.

Comments to the Author:

I appreciate the ability to review this revised manuscript and I continue to agree that the information collected within the current protocol is of particular importance to the field. However, I am concerned that the authors only provided cursory responses to a number of areas of improvement that have been identified during the review process. I understand that the authors are somewhat constrained by the aims that they identified in a grant proposal; however, this should not preclude revisions to better explain their services, the rationale for their proposed analyses, and consideration of additional analyses that would be possible within the constraints of their dataset. From my perspective, such edits will strengthen the reach of the protocol to a broader range of interested parties that may not be as familiar with the specific organization of the clinics and the details of the types of care that they provide.

We would like to thank the reviewer for acknowledging the importance of this study to the field and for the helpful recommendations for revisions to our manuscript.

While the authors provided some explanation about the care pathways present at each clinic within their response to reviewers, this information was not integrated into the manuscript. Given care pathways vary significantly across paediatric gender clinics and the goal of the

protocol is to better understand the profile of CYP progressing through different care pathways, clear description of the care pathways is essential. This includes how families initially make contact with the clinic, what initial visit(s) are scheduled, what subsequent visit(s) are available (including visit types, frequency, and timing/relationship between visits - e.g., are certain visits required, do they occur in a specific progression), the general focus/goal(s) of each visit type, and how recommendations/referrals are made for each visit type. It may be helpful to group appointment types together to make it clearer, i.e., (a) initial/intake appointments, (b) assessment appointments, (b) optional other mental health appointments (consultation, therapy), (c) endocrine care, (d) optional other medical appointments. For VUMC, there are a range of medical appointment types that are unclear, particularly “reading status,” “clinical genetics,” “intercollegial consultation,” “emergency consultation,” “group consultation endocrinology.” It is also not clear why endocrinology is broken into under and over 15 (and in one clinic only) and how an outreach/standard assessment differs from a CYAF assessment.

We thank the reviewer for raising these important issues for clarification. We have now included further information about care pathways and appointment types, in an appendix, which readers can refer to for further detail. See Appendix 1.

With regard to appointment types, the breakdown of appointment types listed in Table 1 (pg. 20-21) reflects all of the possible appointment types that each clinic provides, as recorded on their respective medical records systems, i.e. this information is designed to inform administrative/costing processes, not research. These appointment types will only be used for costing for health economic purposes by the health economist on the research team (Associate Professor Rachael Hunter, University College London). Analysis will focus on attendance to endocrinology appointments, as two aims of the study are to examine the profile and associated costs of CYP who do and do not attend endocrinology appointments. The total number of appointments and wait times in relation to first appointment will also be examined. In line with the reviewer’s suggestion, and, as stated in the manuscript, appointment types will be grouped into ‘assessment’, ‘psychosocial treatment’, ‘group’ and endocrine’ for the purposes of healthcare cost analyses (pg. 11). We have added additional information to Table 1 to indicate which category each appointment type falls into.

The average number of contacts with the service will also be reported for each type of appointment. We anticipate that data pertaining to appointments and healthcare costs will be reported in a separate manuscript led by Associate Professor, Rachael Hunter at UCL.

While the authors focus on identifying characteristics associated with attending endocrine appointments, identifying characteristics associated with attending mental health services could be equally beneficial to improving these services. If sample sizes permit, it would also be very helpful to know if there are any differences between those that remain engaged in care and those that stop attending or are “discharged against professional advice.” Explaining how the latter category is defined would be helpful.

We are grateful to the reviewer for these helpful suggestions. We can confirm that we intend to explore, should sample sizes permit, whether there are any differences between those that remain in care and those that stop attending or are ‘discharged against professional advice.’

The term ‘discharged against professional advice’ refers to disengagement/withdrawal from care against the recommendation of the clinical care team.

Beyond noting that the aim was included in a grant proposal, I still do not see any specific justification for including calculations surrounding the cost of care or information about how this data will be used to improve care. Why were costs associated with care identified as an important research question and why was endocrine care selected in particular (versus all forms of care or mental health services)? Once this data is collected, who will the authors be sharing it with and what are the potential goals and outcomes of sharing the data, including how might they improve care? Given readers interested in better understanding patient demographics and care pathways are likely to be at least somewhat different than those interested in specific cost estimates, it may be advisable to remove cost related aims from the present manuscript. This would also allow room to better specify care pathways and how clinical information will be coded and analyzed.

We would like to thank the reviewer for raising these important issues. We are examining costs associated with care in order to explore how the organisation and resourcing of healthcare might best

serve young people. It is anticipated that the findings will provide information to policy makers and service providers on the costs and benefits of providing GID services within a finite budget and is based on best practice as set out by national decision making bodies such as the National Institute for Health and Care Excellence (NICE) in the UK. Furthermore, the funder of this research is the UK National Institute for Health Research (NIHR) Health Services & Delivery Research programme and a particular interest of this funder is to provide some examination of the health economic impacts of care delivery <https://www.nihr.ac.uk/explore-nihr/funding-programmes/health-services-and-delivery-research.htm>.

Although we have a single protocol for this study, we agree with the reviewer that costs associated with care are likely to be of interest to different readers and we therefore anticipate that these will be reported in a separate manuscript. Our health economist, Associate Professor Rachael Hunter (University College London), will be leading on the health economics paper.

Endocrine care was selected because there has been considerable debate around the use of, and access to, physical/medical treatment pathways for CYP. We hope that the present study will shed light on any potential benefits of endocrine care – it may be that CYP who follow this pathway are less likely to need other types of care, and it is therefore clinically and cost efficient in the long-term. We also hypothesise that there will be differences in endocrine care between the UK and Netherlands, so it is an important variable to include in an analysis looking at the difference in costs between the two countries. Specifying endocrine care is also in line with the other analyses being conducted.

Regarding study aim 5:

(a) The proposed analyses do not match up clearly with the aim. The aim is focused on identifying the cost of care and cost of care is included as one variable potentially predicting who attended an endocrinology visit, but other variables are also included as predictors of attendance.

We apologise that we did not report this clearly: cost of care is the dependent variable. This has now been specified in the manuscript (pg. 11). To further clarify, costs of care will be reported by CYP who attend endocrinology and those who do not. These costs may differ for a variety of reasons including the number of required appointments, duration of time in contact with the service, and the cost of the endocrinology care itself, including medications, clinician time and overheads. The purpose of this analysis is to calculate the cost of care for each CYP from the beginning to end of their time with GIDS,

and then report if different CYP have different costs because (i) of the country they are in; (ii) they followed the endocrine pathway (or didn't); (iii) other clinical factors such as the prevalence of autistic traits. Our current hypothesis is that both country and endocrinology are the two things most likely to influence the cost of care. As a result, we have pre-specified that the results will be reported by these factors.

(b) The authors state that they will include “outcomes including wellbeing (measured by CBCL, YSR and TRF)” as a predictor of attendance. When and how is this longitudinal data collected?

In cases where the CBCL, YSR, TRF, or SRS has been completed multiple times by the same service user, only data from the first time point will be included in analyses of attendance. Such baseline data will also only be included if completed within one year of the referral date, to minimise confounding effects of time and treatment pathways on the scores. For example, a CYP who has waited 14 months might have different baseline values from one who has waited only one month. Furthermore, these outcomes will only be included as predictors of attendance if completion rates are $\geq 70\%$ in the cohort.

(c) “Care pathway” and “potential predictors of costs and outcomes” are not defined.

These terms have now been defined within the manuscript.

“Care pathway” refers to the type of treatment and/or support that the CYP receives throughout their time with the service.

“potential predictors of costs and outcomes” refer to variables such as age at first appointment, gender dysphoria diagnosis, and autistic traits.

What is the relationship between the Adolescent Gender Identity Research group (AGIR) and the Logic study? The authors mention that the clinics are participants in both.

The AGIR is a collaborative research group, established in 2015, that uses the same assessment battery of tests that are both clinically useful and enable cross-clinic research. Both the UK GIDS and the VUMC are part of this group. The LOGIC study is an independent programme of research, funded by the NIHR in the UK and involves a collaboration between the NHS, University College London and the Universities of Liverpool and Cambridge. The LOGIC study does not have a relationship with the AGIR. Further information about the AGIR can be found here:

Dèttore, D., Ristori, J., Antonelli, P., Bandini, E., Fisher, A., Villani, S., & Cohen-Kettenis, P. T. (2015). Gender dysphoria in adolescents: the need for a shared assessment protocol and proposal of the AGIR protocol. *J Psychopathol*, 21, 152-158. <https://www.jpsychopathol.it/article/gender-dysphoria-in-adolescents-the-need-for-a-shared-assessment-protocol-and-proposal-of-the-agir-protocol/>

However, to avoid confusion, reference to the AGIR has been removed from the manuscript because the LOGIC study does not have a relationship with this group (pg. 9).

I am curious about how many service users and organizations provided input into the development of the protocol and whether this included CYP or only parents. Was feedback provided at one point in time or on an ongoing basis?

Thank you for raising this important question. Input was provided by service users and organisations during the funding application process. In addition, the research team includes an academic co-investigator with lived experience of service use. Patient and Public Involvement is also a prominent component of the peer review process for all UK National Institute for Health Research funding applications and there was extensive PPI peer review prior to the grant being awarded. The LOGIC study programme of research has an active study PPI group, comprised of 26 parents/carers and children and young people, who meet 4-6 monthly and provide ongoing input and feedback regarding the conduct of the study and will be involved in providing feedback regarding study findings once available. The Study Steering Group which oversees the conduct and progress of the study meets 4-6 monthly and includes representation from individuals with experience of service use, as well as those with relevant academic expertise independent of the research team. In addition, the LOGIC programme of research includes a prospective mixed methods longitudinal study. The University of Liverpool is leading on LOGIC-Q a longitudinal qualitative study involving children and young people (age 5 to 18 years) and is hearing directly from young people themselves about their experiences of services. We hope in this way to include a range of perspectives and particularly input from children and young people themselves, on an ongoing basis.

Can any of the data being collected within the protocol speak to the potential impact of the Bell vs Tavistock ruling, which severely limits access to gender affirming medical treatment prior to

age 16? This would be of greatest importance and is particularly urgent given the many CYP whose medically necessary treatment is being halted or blocked.

Thank you for highlighting this. The ruling was made in December 2020 which was several months after we submitted this protocol for publication so the ruling itself was not specifically taken into account in the development of this protocol. However we completely agree that this research is now particularly important and urgent.

We would like to thank the reviewer for their thoughtful and helpful review of our paper.